# In Situ Delivery and Production System (*i*DPS) of Anti-Cancer Molecules with Gene-Engineered *Bifidobacterium*

**DOI:** 10.3390/jpm11060566

**Published:** 2021-06-17

**Authors:** Shun’ichiro Taniguchi

**Affiliations:** Department of Hematology and Medical Oncology, Shinshu University School of Medicine, Matsumoto City 390-8621, Japan; stangch@shinshu-u.ac.jp

**Keywords:** solid cancer, microenvironment, hypoxia, cancer therapy, DDS, anaerobic bacteria, *Bifidobacterium*, bacterial therapy, *i*DPS, EPR

## Abstract

To selectively and continuously produce anti-cancer molecules specifically in malignant tumors, we have established an in situ delivery and production system (*i*DPS) with *Bifidobacterium* as a micro-factory of various anti-cancer agents. By focusing on the characteristic hypoxia in cancer tissue for a tumor-specific target, we employed a gene-engineered obligate anaerobic and non-pathogenic bacterium, *Bifidobacterium*, as a tool for systemic drug administration. This review presents and discusses the anti-tumor effects and safety of the *i*DPS production of numerous anti-cancer molecules and addresses the problems to be improved by directing attention mainly to the hallmark vasculature and so-called enhanced permeability and retention effect of tumors.

## 1. Introduction

### 1.1. Molecular Target Cancer Therapy and its Limitations

One of the greatest advances in recent cancer research is the identification of driving genes, which are specific to cancer type and critically responsible for cellular growth. Many molecular targeting drugs have now been developed, leading to increased specificity to cancers and fewer side effects on bone marrow and digestive organs, the most common areas harmed by classic chemotherapeutic anti-cancer drugs [1]. However, there remain obstacles in cancer treatment, such as the appearance of drug-resistant cells from heterogeneous cancer cell populations, which leads to recurrence. There are also new types of side effects differing qualitatively from those of conventional cytotoxic anti-cancer drugs [2]. In the case of solid cancers, simple but troublesome problems exist in that the exposed dose of drugs is often insufficient to kill cancer cells as compared with hematopoietic cancers, which are more readily exposed to anti-cancer drugs. Accordingly, it is essential to develop a selective drug administration system to deliver large amounts of anti-cancer drugs to solid tumors and overcome the situation.

In their review on the hallmarks of recent cancer research leading to the identification of driving genes and development of molecular targeting drugs and antibody drugs, Hanahan and Weinberg pointed out the importance of the characteristic microenvironment of cancer, including low oxygen pressure (pO_2_) and immune avoidance conditions [3], as targets for emerging therapies [4]. Thus, it may be desirable to focus on the tumor microenvironment rather than attack individual cancers, which contain heterogenous populations of cells [5] that can produce drug resistance.

### 1.2. The Enhanced Permeability and Retention (EPR) Effect

To overcome the above difficulties, it was suggested that anti-cancer drugs of a high molecular weight should be employed to make use of the characteristic vasculature of tumor tissues [6]. In malignant tumors, leaky vasculature with 100–1000 nm pores is generally formed due to the rapid but immature formation of vessels in association with cancer growth. In addition to fragile blood vessels [7,8], there is usually poor lymphatic drainage, resulting in the retention of large molecules in tumor tissue. This phenomenon was discovered by Maeda, who named it the EPR effect. By focusing on EPR, many trials have managed to successfully target tumor tissues [9].

### 1.3. Our Approach Targeting the Low pO_2_ in Solid Cancers with Bifidobacterium

We have been working to establish a system for the selective and continuous production of anti-cancer molecules in tumors [10,11]. For this purpose, we directed our attention to the hypoxic conditions in solid cancers for a therapeutic target. As a tool for the local production of anti-cancer drugs, we adopted the obligate anaerobic and non-pathogenic bacterium, *Bifidobacterium*. Our first paper in 1980 showed the selective growth of *Bifidobacterium* in the cancer tissues of tumor-bearing mice where the bacilli were intravenously injected [12]. Currently, several gene-engineered *Bifidobacterium* lines have been established to produce numerous anti-cancer molecules in a process we have termed the in situ delivery and production system (*i*DPS). A new anti-tumor drug made with the *i*DPS is now undergoing clinical testing.

### 1.4. Hypoxia and Immature Blood Vasculature in Malignant Tumors

Tumor hypoxia is a well-known phenomenon [13,14,15]. The median pO_2_ in tumor tissues is lower than that in normal tissues, which is never below 10 mm Hg. Hypoxia is generally observed in tumor tissues in spite of active angiogenesis. This paradoxical phenomenon has been attributed to impaired vascular communication and networks leading to functional, but chaotic, shunting and dysfunctional microcirculation [16]. Thus, even in the presence of blood vessels carrying fresh oxygen and nutrients, shunts to other blood vessels form easily, such that the downstream vessel will be not supplied, leading to hypoxia and/or necrosis [16].

Our project idea of the *i*DPS originally derived from Malmgren’s work of injecting anaerobic *Clostridium tetani* spores into animals [17]. In his experiments, tumor-bearing mice died of tetanus due to the strongly toxic neuro-active substances produced by germinated *Clostridium tetani*, while normal mice survived. This was a strong piece of biological evidence for hypoxia existing in tumors; the spores of the anaerobic bacteria had germinated and produced strong toxins in the hypoxic conditions of the tumors, and the highly toxic poison leaked from the tumor tissues to kill the host despite very small amounts. This led us to the idea of non-pathogenic anaerobic bacteria as a tool for safely and selectively targeting solid cancers while sparing the host. 

### 1.5. Bacterial Therapy for Cancer

Bacterial cancer therapy has a long history. The accidental tumor regression by clostridial infection clinically observed by Vautier in 1813 [18] launched a number of bacterial therapy experiments. Later, the recovery of a patient with inoperable lymphosarcoma by erysipelas prompted Coley to begin treating cancer patients with live erysipelas agents and/or bacterial toxins [19]. However, ensuing trials were limited, likely due to the difficulty in controlling toxicity and a shift to chemotherapy and radiation treatment. Recently, however, bacterial therapy has been revived with the use of genetic manipulation and is a promising method for cancer therapy [20,21,22,23]. 

Nowadays, several clinical studies on bacterial cancer therapies, including our own, are underway in the United States. In most cases, *Salmonella* or *Clostridium* is used. The earliest clinical trials approved by the American Food and Drug Administration (FDA) were carried out by Rosenberg’s group at the National Institutes of Health (NIH) using *Salmonella* [24] and by a group at Johns Hopkins University with *Clostridium* [25]. Other trials have produced anti-tumor responses in both animals and human clinical studies [26,27,28]. In the above cases, the bacteria are attenuated due to their highly virulent nature. Despite concerns on the appearance of revertants and whether facultative anaerobic bacteria can be completely removed from normal aerobic tissues and cells, recent clinical trials have largely managed to maintain host safety [23]. 

#### The EPR Effect’s Importance in Bacterial Therapy

Maeda’s group noted that though EPR effect is applicable to particles of μm size (i.e., bacteria) or macromolecules of ∼1000 kDa, nanocarriers with diameters of ∼100 nm are known to achieve better or optimal EPR-based tumor accumulation [29]. Thus, the accumulation of even aerobic bacteria in tumors may be explained by the EPR effect [30]. Importantly, this effect can be augmented by vascular dynamic modifiers, such as nitroglycerin, from which nitric oxide is produced in the hypoxic conditions of tumors. In his attempts to target tumors with lactobacillus, Maeda’s group showed that the number of bacteria localized in tumors increased by ten-fold as compared with controls [30].

## 2. Our Trials for Bacterial Therapy 

### 2.1. Selective Localization of i.v. Injected Bifidobacterium in Tumors

Figure 1 shows our first data reported in 1980 [12] on the growth of i.v. injected *Bifidobacterium* in tumors (Figure 1a) along with the results of our genetically engineered *Bifidobacterium* (Figure 1b) (Farumashia,15, (5), 438–440, 2015 [in Japanese]). In both cases, *Bifidobacterium* selectively grew in tumor tissues and became rapidly diminished in the blood and normal tissues, including the relatively hypoxic bone marrow known as a niche for hematopoietic stem cells. No acute toxicity was observed, and the survival of mice i.v. injected with *Bifidobacterium* was comparable with that of control animals, demonstrating the absence of chronic toxicity [12]. 

It was noteworthy that the survival of the normal animals in Malmgren’s experiments [17] indicated that the *Clostridium* spores did not geminate in the bone marrow, thus demonstrating that bone marrow pO_2_ was insufficiently low for obligate anaerobic bacteria germination and/or an inadequate EPR effect to trap the spores or bacteria. This phenomenon also likely occurred in our experiments. 

### 2.2. Transformation of Bifidobacterium with an Expression Vector for Cytosine Deaminase (CD)

Although we initially sought to transform the bacteria to produce anti-cancer molecules, no plasmid was available for *Bifidobacterium* in the 1980s. In 1997, however, an expression vector developed by Kano’s group [31] launched a series of collaborative trials for the creation of anti-cancer drugs by *Bifidobacterium*. We first inserted the CD gene of *E. coli* into the vector. The CD enzyme can convert the low-toxic 5FC, a prodrug of 5FU, to the toxic anti-cancer drug 5FU. 5FC is a well-known drug for mycosis, with almost no systemic toxicity by oral administration. We transformed *Bifidobacterium* with the CD expression vector and began experiments on solid cancers using genetically engineered *Bifidobacterium* in combination with 5FC [32,33,34,35]. This was the first step towards our *i*DPS.

### 2.3. Therapy Experiments on Solid Cancers Using Transformed Bifidobacterium with 5FC

The procedure for our cancer treatment with *Bifidobacterium* carrying the CD gene is as follows [11,36]. First, we *i.v.* injected the transformed *Bifidobacterium* into tumor-bearing animals. Several days later when the specific localization of the bacteria inside the tumor tissues was expected, we commenced oral 5FC administration to the animals. Although the prodrug spread throughout the body, it was converted to 5FU only in tumor tissues by *Bifidobacterium* expressing CD. We then checked for tumor growth suppression and systemic toxicity by 5FU. In the first experiment, we used an autochthonous DMBA-induced rat breast cancer system, and comparable results were obtained in various tumor-bearing animals. The selective localization of *Bifidobacterium* in tumors was confirmed by tissue homogenate cultures in vitro and Gram-positive staining of *Bifidobacterium* in the tumor tissues. CD expression in *Bifidobacterium* was ascertained by immunostaining with anti-CD antibodies.

The success of our therapy system, i.e., the suppression of tumor growth in chemically induced autochthonous rat breast cancer, can be seen in Figure 2a. When we treated human breast carcinoma transplanted into immunodeficient nude mice, tumor suppression was again witnessed without systemic toxicity (Figure 2b). Indeed, relatively large 5FU production was detected exclusively in tumors and not in normal tissues (Figure 2c). An important point was that no apparent adverse effects were observed, which was also the case in dogs, monkeys, and other large animal tests. 

We later sought to increase enzyme activity by modifying the active site of CD according to Mahan’s method [37]. Since the original substrate for CD is not 5FC, but rather cytosine, the enzymatic activity and affinity to 5FC was relatively low as compared with that to cytosine. When we changed the amino acid at position 314 of the active center of the enzyme from aspartic acid to alanine, the conversion rate of 5FC to 5FU was increased by approximately ten-fold. In clinical trials, the modified expression vector was further tailored by removing the resistance gene to an antibiotic, spectinomycin, to protect against horizontal transmission.

### 2.4. Immunological Safety

To test for immunological toxicity and possible severe anaphylaxis from repeated injection of the bacteria into animals, we evaluated for active systemic anaphylaxis (ASA) reactions and passive cutaneous anaphylaxis (PCA) caused by IgG with a sensitive guinea pig system (Table 1). In terms of ASA reactions, the positive control ovalbumin induced severe shock, whereas little, if any, reactions were seen for *Bifidobacterium*. Regarding PCA, although IgG antibody formation against *Bifidobacterium* had been suggested, experiments using animals immunized with *Bifidobacterium* confirmed the safety and therapeutic efficiency of the system. 

We further examined for the induction of inflammatory cytokines related to *Bifidobacterium* in collaboration with an expert of infectious immunity, Tsutsui, Hyogo College of Medicine in Japan. As shown in Figure 3, the *E. coli* control predictably induced the cytokines seen in sepsis, while *Bifidobacterium* did little [12]. Since cytokine production occurs through Toll-like receptors, which are the main players in innate immunity, those results suggested that *Bifidobacterium* was recognized neither by Toll-like receptors in vivo [38], nor by other innate immunity systems activating IL-1β through inflammasomes in the cytosol [39]. It is well known that inflammasomes are activated by the flagella of *Salmonella* to induce IL-1β.

Concerning the above findings, an interesting report found that extracellular vesicles in the blood inhibited the induction of NF*κ*B expression by *Bifidobacterium* [40]. *NFκB* is a well-known master gene for inflammatory cytokines. When *Bifidobacterium* was allowed to act on cultured human embryonic kidney cells, NF*κ*B was induced in the absence of serum. However, serum addition to the medium suppressed NF*κ*B expression in a concentration-dependent manner. It was also shown that extracellular vesicles in the serum played a key role in suppressing NF*κ*B. Those results suggested that the lack, if any, little of inflammatory cytokine induction in mice by *Bifidobacterium* was partially attributed to extracellular vesicles in the blood.

In the field of probiotic research, human intestinal flora changes have been examined for relationships between flora variety and health conditions. Shortly after birth, *Bifidobacterium* becomes the main gut flora and coexists with the organism throughout life at gradually decreasing amounts [41,42]. It is likely that humans and other mammals have some immune tolerance against *Bifidobacterium*. These facts strengthen our notion that *Bifidobacterium* can be safely used for an *i*DPS to produce anti-cancer molecules in humans. Moreover, *Bifidobacterium* has been included in commensal microbiome work to enhance the cancer therapy efficiency of anti-PD-1 antibodies [43].

### 2.5. Translational Research

For the purpose of applying *i*DPS/5FC for clinical cancer treatment in humans, rigorous testing was performed according to Chemistry, Manufacturing and Control (CMC) and Good Manufacturing Practice (GMP) guidelines. CMC requires showing the physicochemical properties of the product in detail, while GMP necessitates the provision of quality assurance that products are consistently produced and controlled to quality standards. Since CMC and GMP are generally difficult processes even with simple chemical compounds, certification for living bacteria has been much more challenging.

We have performed the precise characterization of various aspects of *Bifidobacterium*, including its membrane composition, stability of the expression vector in which the antibiotic resistance gene for selective pressure and other sequences were removed for safety, and bacterial survival measurement methods. Every attempt has been made to establish virus-free and sterile preparations for GMP. Finally, through investigational new drug discussion, our proposed first-in-man clinical trial was approved by the American FDA, with NIH Recombinant DNA Advisory Committee approval of our protocol on biosafety. The first-in-man phase 1 and 2 tests were approved in 2013 and carried out sponsored by a bio-venture company, Anaeropharma Science Inc., Tokyo, Japan. Whereas the phase 1 test is almost completed, phase 2 has regrettably been postponed by the current COVID-19 pandemic.

The concept for applying the *i*DPS using genetically modified *Bifidobacterium* on human cancer therapy is much the same as that with animals. Since the line of CD-expressing *Bifidobacterium* for application on humans is named APS001F, the clinical trial has been entitled “A Phase I/II Safety, Pharmacokinetic, and Pharmacodynamic Study of APS001F with Flucytosine (5FC) and Maltose for the Treatment of Advanced and/or Metastatic Solid Tumors” [44]. 

### 2.6. Combination Therapy of APS001F Plus 5FC (APS001F/5FC) in Combination with the Immune Checkpoint Inhibitor (ICPI) Anti-PD-1

With recent developments in cancer treatment, the prominent anti-tumor effects of ICPIs, such as anti-CTLA4 and anti-PD-1 antibodies, have been demonstrated worldwide [45,46,47]. To augment the effects of ICPIs, combination treatments with chemotherapy, molecular targeting anti-cancer agents, and/or radiation therapy have been tested. However, the associated side effects often increased in tandem with anti-tumor action. We expected our *i*DPS using *Bifidobacterium* to enhance ICPI treatment by 5FU in tumors without raising side effects by enhancing immune reactions through innate immunity locally stimulated by *Bifidobacterium*. 

Combination therapy experiments of the *i*DPS with APS001F/5FC and anti-mPD-1 antibodies have already yielded promising results [48]. The almost completed clinical phase 1 test of APS001F/5FC also serves for prechecking whether the combination of anti-PD-1 antibodies, which are already available for clinical use, with APS001F/5FC can be a potential treatment candidate. The rationality for this next step was made through investigation of the literature [49,50,51,52] as follows: (1) combining ICPIs and anti-cancer drugs, including 5FU, reportedly enhances anti-tumor effects, (2) 5FU has the potential to suppress myeloid-derived suppressor cell inhibition of anti-tumor immune reactions, and (3) the systemic administration of 5FU at a high dose rather impairs anti-tumor immunity. The third consideration may be attributed to the systemic toxicity of 5FU, which is improved by the *i*DPS with APS001F/5FC. 

Combining APS001F/5FC with anti-PD-1 antibodies in therapeutic experiments enhanced treatment effects (Figure 4a) without increasing side effects. We first observed that the tumor growth in animals treated with anti-PD-1 antibodies was slightly suppressed. When combined with APS001F/5FC, however, this effect was significantly augmented. Animal survival was also prolonged, including a complete remission case [48]. 

By analyzing the immune cells in tumors during combination therapy, we witnessed a remarkable decrease in regulatory T (Treg) cells, which inhibit anti-tumor immune activity, and consequently the ratio of CD8 cells to Treg cells was greatly increased (Figure 4b) through a yet unspecified mechanism. In this therapy system, 5FU was produced in situ in tumor tissues and Treg cells were suppressed without a change in CD8 activity, likely raising the anti-tumor effect of combination therapy. We strongly believe that APS001F/5FC in future clinical trials will exert promising effects in combination with anti-PD-1 antibodies and other ICPIs. Furthermore, combined treatment with anti-tumor drugs and ICPIs may be possible with a single gene-engineered *Bifidobacterium* clone. Although technically challenging, we have already succeeded in establishing such a co-expression system with *Bifidobacterium.*

### 2.7. Establishment of a Protein-Secreting System 

We are underway to engineer *Bifidobacterium* that not only express, but also secrete, proteins such as anti-tumor antibodies and cytokines (Scheme 1). In spite of the prominent effects of ICPIs, there remain problems including autoimmune diseases and other severe side effects. Immune checkpoint molecules, such as CTLA4 and PD-1, inactivate T-cell killing to terminate excessive inflammatory reactions in the body. It is physiologically important to halt unnecessary immunoreactions, and blocking those molecules tends to induce immune toxicities in the host [53,54,55].

To improve this situation, it will be useful to establish an *i*DPS for immune checkpoint-modifying antibodies, including anti-CTLA4 and anti-PD-1 antibodies, and immune modifiers, such as anti-tumor cytokines, with *Bifidobacterium.* We have therefore been attempting to establish *Bifidobacterium* that both express and secrete immunological anti-cancer molecules (Scheme 1), such as anti-CTLA4 and/or anti-PD-1 antibodies, in addition to such immune-stimulating anti-tumor cytokines as TNFα and INFγ. 

#### 2.7.1. Anti-HER2 scFv

As it is generally difficult to produce the original structure of antibodies with bacteria, we firstly tried to create single-chain variable fragments (scFvs), which were fusion proteins of variable light and heavy chains connected with a linker for each antibody, such that the scFv could be called a single-chain antibody. 

In our attempts to produce scFvs for immune checkpoint molecules, we started by expressing and secreting a biologically active scFv already made and confirmed by another system. For this purpose, we turned to an anti-HER2 antibody, trastuzumab, since a biologically active scFv for trastuzumab had been developed by Akiyama at the Shizuoka Cancer Center in Japan. Trastuzumab is well known and widely used as a molecular targeting antibody for human breast cancer, but occasionally induces cardiotoxicities [56,57]. Thus, we sought to establish *Bifidobacterium* secretion of the biologically active scFv for anti-HER2 antibodies. 

We succeeded in making not only an expression, but also a secretion, system for a biologically active scFv derived from the anti-HER2 trastuzumab with *Bifidobacterium* (Figure 5) [58]. Production of the scFv to human HER2 was confirmed by Western blot analysis. We also verified biochemical activity by FACS and immunological staining [58]. The genetically modified bacteria were injected into nude mice bearing the human HER2-positive breast cancer, NCI-N87. We witnessed selective localization of the bacteria inside the tumor, secretion of anti-HER2 single-chain antibodies, and ultimately a suppressive effect on tumor growth (Figure 5) [58]. This success in creating *Bifidobacterium* to express and secrete the biologically active trastuzumab scFv prompted us to establish *i*DPS with *Bifidobacterium* for ICPIs.

#### 2.7.2. scFvs for ICP Antagonistic or Agonistic Antibodies, including Anti-PD-1, Anti-CTLA4, Anti-41BB, and Anti-Tumor Cytokines

We next established *Bifidobacterium* to secrete scFvs of anti-PD-1 [59], anti-CTLA4 [60], and anti-41BB (an immune checkpoint agonist) [61] antibodies. All scFvs produced by the *i*DPS were detected exclusively in tumor tissues and exhibited immunological activity and anti-tumor effects without remarkable side effects. 

In addition to scFvs, we have also established *Bifidobacterium* expressing and secreting INFγ and/or TNFα, which displayed notable anti-tumor effects [62,63]. It is well known that the clinical application of INFγ and TNFα is difficult due to their severe systemic toxicity in spite of strong anti-tumor effects [64]. However, this was not the case in our *i*DPS using *Bifidobacterium*. When we systemically *i.v.* injected *Bifidobacterium* that could secrete INFγ and/or TNFα into tumor-bearing animals, high amounts of cytokines were detected in tumors, with little in the blood and thus no systemic toxicity. In addition to the anti-tumor properties of *Bifidobacterium* expressing and secreting the cytokines, the enhancement of anti-tumor effects by combination with ICPI antibodies or a popular anti-cancer drug, Adriamycin, was observed as well. Taken together, drugs that have been unsuitable for clinical use due to strong systemic toxicity despite formidable anti-cancer properties may be revisited through the use of our *i*DPS. Most recently, a *Bifidobacterium* clone, APS002, has been established to secrete diabodies against EGFR/HER3 and CD3 and redirect T cells to EGFR/HER3-positive cancer cells. This clone inhibited the growth of human EGFR-positive cancer cells transplanted into humanized immunodeficient mice [65], indicating a possible clinical application of this engineered *Bifidobacterium*. 

## 3. For Further Improvement of the *i*DPS

### 3.1. Notes for the Presnt iDPS 

#### 3.1.1. Animal Experiments

The established methods for *i*DPS so far are detailed in our previous papers, especially in [11] and patent information [63]. In animal experiments with *Bifidobacteria*, we used chemically induced autochthonous tumor system, mouse tumor model, and allogenic transplanted human tumors in immunodeficient mice (Figure 1 and Figure 2). Generally, autochthonous cancer is relatively difficult to cure compared with a transplanted tumor, because the tumor is comprised of cells which have escaped from host immune surveillance. In this sense, it is thought to mimic the human cancer system. In transplantation of cancer cells, the number of cells should be as small as possible to mimic a human tumor system where one nodule is produced from single or a few cancer cells. In such systems, we have experienced that *Bifidobacteria* tended to be localized even in small tumors. While *Bifidobacterium* could be safely administered to immunodeficient mice, immunodeficient mice are always infected with various bacteria as compared with normal mice because they are immunodeficient. Thus, during the assay of the number of *Bifidobacteria* in various tissues, it was required to remove intrinsically contaminated germs in in vitro bacterial culture system. Our genetically engineered *Bifidobacteria* have 5FU resistance in addition to spectinomycin, so that we were able to eliminate such germs and to identify the colony of *Bifidobacterium* by using both drugs in vitro bacterial culture. 

In the assay for inflammatory cytokines, there was little or no induction of such cytokines in our mouse system (Figure 3), however, it probably depends on the type of animal, including human. More detailed analysis is needed from the viewpoint of molecular immunology. 

In our present chemotherapy system (Figure 2), 5FC have been used as the prodrug. Since a *Bifidobacterium* clone expressing β-glucuronidase has also been established to activate prodrugs inactivated by glucuronic acid conjugation. Such clone will be useful to reuse drugs that have strong anti-tumor properties but severe systemic side effects. 

In the protein secretory system (Figure 5), the optimum secretory signal peptide depended on the secretory protein. We have searched various secretory signals of the *Bifidobacteriu*m’s own secretory proteins and adequate promoter for the gene as well. In order to improve combination therapy, we have established various vectors to co-express/secrete anti-tumor cytokines and/or scFvs for various antitumor antibodies, which will be useful for combination therapies of cancer in the future.

At the preclinical level, there are some studies using *Bifidobacteria,* though there seems to be only our group that has advanced to clinical trials. One of the preclinical studies similar to ours demonstrated the successful delivery and efficient expression of Tumstatin, a powerful angiostatin, with genetically engineered *Bifidobacterium*, leading to antitumor effects through inducing apoptosis of tumorous vascular endothelial cells [66]. They observed that *Bifidobacterium longum*, selectively localizes to and proliferates in the hypoxia location within solid tumor. The other investigated the therapeutic effect of new recombinant *Bifidobacterium breve* strain expressing interleukin (IL)-24 gene on head and neck tumor xenograft in mice and reported that new recombinant bacterium has the capability of targeting tumor tissue *in vivo* [67]. These reports are consistent with our results in terms of selective localization of *Bifidobacterium* in tumors.

#### 3.1.2. As for Safety of *i*DPS with *Bifidobacterium*

For the clinical trial, bacteria dosage was initiated at less than a NOAEL (no observed adverse effect level) in the most sensitive dogs, and the dose was gradually increased with the minimum homing dose for rat transplanted cancer as a guide [44].

Since the *Bifidobacteri**um* we use is derived from enterobacteria in human, which is also used as intestinal regulators and yogurt as a probiotics, and thus it has been considered to be a priori non-toxic from experience. However, to evaluate the toxicity of genetically modified *B. longum*, a number of preclinical studies have been also carried out in several animal species, including normal mice, nude mice, normal rats, nude rats, dogs and monkeys. Both pharmacological and preliminary general toxicity studies were done, none of which revealed serious unfavorable toxicities [10]. Various antibiotics were examined to eliminate excess *Bifidobacteria* after treatment, and many antibiotics were found to be effective. In particular, we have confirmed that it can be easily removed with commonly used penicillin antibiotics (data, not shown).

#### 3.1.3. The Specific Advantages of Using Bifidobacterium for Our *i*DPS

The specific advantages of using *Bifidobacterium* and the reason why we have been using *Bifidobacterium longum* for our *i*DPS are as follows: (1) It is an obligate anaerobic bacterium, so that it can discriminate hypoxic malignant tumor tissues for the colony formation from normal tissues. (2) It does neither produce toxic substances, nor has flagella which activate inflammasome to induce IL-1β. (3) It has been generally regarded as a good bacterium derived from human intestinal bacteria, so that it is easy to think about safety a priori even if it is administered into the blood, thus leading to a sense of security for the recipient. In addition, (4) it has also been shown to work positively in treatment with antitumor immune check point inhibitor, anti-PD-1 antibody. 

Since an expression vector has become available firstly for *Bifidobacterium longum*, this *Bifidobacterium longum* has been used as a tool for *i*DPS and we came to apply for IND with *longum*. However, if safety is ensured by other probiotic species and the use of expression vectors becomes possible for them, they could become a better tool, so that it seems important to continue to search such species in the future.

### 3.2. Seeking an Ideal Micro-Factory with Guaranteed Safety 

Nowadays, bacteria can be modified to endow new phenotypes using gene engineering [18,23]. It will be important to improve the efficiency of *i*DPSs by modifying *Bifidobacterium* to develop ideal delivery and in situ production systems. The success and safety of *i.v.* administration of APS001F with living *Bifidobacterium* in our clinical phase 1 trial was an encouraging first step. Based on the clinically confirmed safety of systemic *Bifidobacterium* injection, we will next modify *i*DPS setups to produce an ideal micro-factory of various anti-cancer molecules for selectively and continuously treating all types of solid cancer.

To strengthen the concept of *i*DPS tolerance in clinical applications, its molecular safety mechanism needs deeper understanding from the viewpoint of the innate and acquired immunological reactions to *i.v.* injected *Bifidobacterium.* We believe that we can ameliorate our system towards completely and safely eradicating cancer.

There are several issues to consider when improving and strengthening *i*DPS micro-factories in tumors. First, it will be necessary to increase the number of bacteria in tumors to provide clear therapeutic effects. One way is the simple quantitative approach of increasing the inoculation size as we have not yet determined the tolerable maximal dose. Concerning qualitative modifications, it will be critical to consider two main factors: (1) The dynamics of tumoral blood flow preventing bacteria entrance into the tumor, and (2) The capture of bacteria by reticuloendothelial cells and neutrophils that rapidly reduces bacterial density in the blood.

#### 3.2.1. Modification of Tumor Hemodynamics with Vasodilators to Enhance the EPR Effect

In our research, the number of bacteria detected in each tumor-bearing animal varied and depended on the tumor type. In order to consistently obtain a large number of bacteria in any type of tumor, we will need to consider modifying the hemodynamics of lesions by directing attention to the EPR effect [9].

Inside the tumor, it is well known that blood vessels occasionally become blocked by poor blood flow dynamics, which may hinder the entrance of *Bifidobacterium* and other macromolecules. In future clinical trials, transiently exposing tumor blood vessels to vasodilators, such as nitroglycerin, may contribute to improved *i*DPSs. Since nitroglycerin reportedly augments the anti-tumor effects of chemotherapies [68,69,70,71], it seems logical to use this agent to enhance the accumulation of *Bifidobacterium* in tumors by increasing the EPR effect. Maeda’s group targeted cancers with lactobacillus in animal experiments in combination with nitroglycerin and showed that the number of bacteria localized in tumors increased by ten-fold versus controls [30]. 

As an additional factor related to poor blood flow in tumors, we may have to consider thrombus inhibition of the intra-tumoral accessibility of anti-cancer drugs. Thrombosis can occur in cancer patients [72,73,74], in whom blood clots may form easily in tumor tissues [75]. The local administration of recombinant plasminogen activators may be effective to dissolve such clots [76]. If bacteria are equipped with such an enzyme by gene-engineering, their accumulation in tumors will likely become enhanced. 

Another factor to potentially improve the bacterial accumulation is to make the bacteria smaller. In a liposome study of pancreatic cancer, smaller liposomes could deliver greater amounts of anti-cancer drugs to the lesion and augment therapeutic effects [77]. Tunability of bacterial size has been investigated [78]. Interestingly, it was reported that deletion of the *Bacillus subtilis ponA* gene encoding for PBP1 (a class A penicillin-binding protein), a bifunctional peptidoglycan synthase, led to thinner cells [79], indicating that it may be possible to change the size of the bacteria by gene engineering. 

#### 3.2.2. Transient Evasion from Bacteria Trapping by the Reticuloendothelial System (RES) and/or Neutrophils

Clinical trials using bacteria for cancer treatment have found that the amount of bacteria accumulation in tumors appears to be less than expected based on the results of animal experiments [23]. 

One reason may be a more rapid capture of injected bacteria by the RES and/or neutrophils than in animal trials. A similar phenomenon is seen for liposome-type drugs. The RES in the human liver and spleen is a major obstacle to the tumor delivery of macromolecular drugs and liposomes [80,81], the effect of which seems to be stronger than in animal systems. Therefore, higher doses may be needed to achieve satisfactory therapeutic effects in humans along with a method to temporarily avoid trapping by the RES.

In one study, covering liposomes with polyethylene glycol (PEG) by so-called PEGylation enabled a breakthrough in the field of liposomes to avoid RES trapping. PEGylation has been also attempted at the cellular level with promising results [82,83]. Additional trials may bring about the same and/or improved effects as PEGylation. In that way, the removal of membrane molecules on bacteria responsible for phagocytosis by the RES and/or neutrophils by gene-targeting [84] will help avoid trapping by the RES. Furthermore, the genes encoding the antigens recognized by reticulocyte-endothelial cells and neutrophils can be replaced with ones produce anti-cancer molecules. As a result, the bacteria will evade detection by the RES and neutrophils, leading not only to an increase in bacterial number entering the tumor microcirculation, but also enabling the bacteria to more stably express anti-cancer molecules. 

In addition to genome editing, a sophisticated method to control plasmid copy number has been reported [85], which will be a powerful tool for enhancing bacterial cancer therapy in the future.

#### 3.2.3. Other Factors for Improving the *i*DPS

Regarding other factors contributing to the improvement of *i*DPSs, bacterial proteolytic activity [86] might be able to widen the localization area of bacteria, thus also being effective to spread anti-cancer substances produced by bacterial micro-factories to the whole tumor region from the central necrosis and/or periphery of the necrotic region where the anaerobic bacteria are colonized. For better distribution of anti-cancer substances in tumor tissue, there may be other ways to conjugate anti-tumor substances with oligopeptides to penetrate tumor masses and consequently widen the diffusion area of therapeutic agents [87].

It is also possible that specific and effective energy sources or nutrients that selectively stimulate the growth of bacteria in solid cancers can increase the number of intra-tumoral bacteria, even if the initial localization number is small. In the case of *Bifidobacterium*, lactulose was firstly used to enhance bacterial number in animal experiments. Lactulose is a disaccharide made up of galactose and fructose that cannot be used by mammalian cells as an energy source. Although considered an ideal bacterial energy source, the *i.v.* administration of lactulose is not permitted in the clinical setting. Therefore, maltose has been used in phase 1 testing as an alternative energy source. The search for an ideal nutrient in clinical trials continues for *Bifidobacterium* [88]. If lactic acid can be used as an energy source, it will be abundantly supplied by the tumor as a metabolite of cancer cell glycolysis. Other bacteria, such as *Veillonella*, consume lactic acid as reported in a meta-omics analysis of elite athletes as a performance-enhancing microbe that functions via lactate metabolism [89]. Transferring this metabolic phenotype to *Bifidobacterium* through gene transfer techniques may create a better micro-factory for anti-cancer drug production. 

## 4. Conclusions

The present review of our novel *i*DPS with *Bifidobacterium* demonstrates its strong potential to safely improve the current problems of solid cancer treatment. Further advances will lower medical expenses through the continuous production of anti-cancer agents and cost-effectively reintroduce discontinued drugs that have strong anti-tumor properties but severe systemic side effects. 

With a concerted global effort, it will be possible to pursue and realize an ideal bacterial-based system, no matter what difficulties await; all that is needed is to remove the unnecessary and undesirable genes and replace them with beneficial ones. Our novel *i*DPS with *Bifidobacterium* represents a promising therapeutic candidate for solid tumors as an in situ self-propagating micro-factory.

## Data Availability

Not applicable.

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
