# Peer review of "In Situ Delivery and Production System (iDPS) of Anti-Cancer Molecules with Gene-Engineered Bifidobacterium"

_jpm, 2021, doi:10.3390/jpm11060566_

Round 1
Reviewer 1 Report
In this manuscript, the authors reviewed an in situ delivery and production system (iDPS) developed previously by the authors for systemic anti-cancer molecules administration in different aspects including background, history, safety, effects, improvement, and future directions, etc in details. Overall, the manuscript is well written, well organized and comprehensively described, which provides adequate information on the topic. If the authors could add some more information regarding the specific advantages of choosing Bifidobacterium as the tool, but not other many potential probiotic species, that would be great.
Author Response
Please see the attachment.
Reply to Reviewer 1Thank you for your taking time to review my manuscript. I tried to revise my paper. I hope you are satisfied with my revision. According to your advice, I added the following content to the Discussion part. Please see the line 544-558 in the revised paper
3.1.3. The specific advantages of using Bifidobacterium for our iDPS
The specific advantages of using Bifidobacterium and reason why we have been used Bifidobacterium longum for our iDPS are as follows. 1) It is an obligate anaerobic bacterium, so that it can discriminate hypoxic malignant tumor tissues for the colony formation from normal tissues. 2) It does neither produce toxic substances, nor has flagella which activate inflammasome to induce IL-1b. 3) It has been generally regarded as a good bacterium derived from human intestinal bacteria, so that it is easy to think about safety a priori even if it is administered into the blood, thus leading to a sense of security for the recipient. In addition, 4) it has also been shown to work positively in treatment with antitumor immune check point inhibitor, anti-PD-1 antibody. Since an expression vector has become available firstly for Bifidobacterium longum, this Bifidobacterium longum has been used as a tool for iDPS and we came to apply for IND with longum. However, if safety is ensured by other probiotic species and the use of expression vectors becomes possible for them, they could become a better tool, so that it seems important to continue to search such species in the future.

Reviewer 2 Report
The review article titled "In Situ Delivery and Production System (iDPS) of Anti-Cancer 2 Molecules with Gene-Engineered Bifidobacterium" discusses about a unique method of bacterial engineering as a drug delivery mechanism to target cancers.
Although written as a review article, this may not fall under its notion as it seems to be a compilation of authors' research. However, the reviewer acknowledges the substantial contribution made by author over the years in furthering the field of in situ drug delivery mechanisms.
The reviewers' major concern is technical in nature.
The figures presented have already been previously published in peer reviewed journals.
Minor concerns.
- Figure 1, 2 and 3 require re-formatting and better description in legends.
- If the results are to be discussed a methods section shall be helpful.
Thanks
Author Response
1)The review article titled "In Situ Delivery and Production System (iDPS) of Anti-Cancer 2 Molecules with Gene-Engineered Bifidobacterium" discusses about a unique method of bacterial engineering as a drug delivery mechanism to target cancers. Although written as a review article, this may not fall under its notion as it seems to be a compilation of authors' research. However, the reviewer acknowledges the substantial contribution made by author over the years in furthering the field of in situ drug delivery mechanisms.
2)The reviewers' major concern is technical in nature.
3)The figures presented have already been previously published in peer reviewed journals.
4)Minor concerns.
- Figures 1, 2 and 3 require re-formatting and better description in legends.
- If the results are to be discussed a methods section shall be helpful.
Reply to Reviewer 2
Thank you for your careful review. I hope that my revision will be satisfactory to you.1) This paper was written based on a request from the guest editor Dr. Hiroshi Maeda to write a kind of compilation on the iDPS I have been researching. So that, it may a little deviate from the usual review paper format. I would like to thank you for acknowledging the content despite the irregular formats.2)&4) 2. According to your comments, I added notes to provide information concerning experimental techniques of iDPS. Please see the line 485-518 in the revised paper. For more technical details, I would like you to see the cited our papers [11] on Methods and protocols of iDPS and [63] on patent information, and other our papers as well.
3) &4)1.
According to your advice, Fig.1-3 and Table 1 have been re-formatted with some revision in the legends.

Reviewer 3 Report
The authors basically reviewed their own work on the iDPS with gene-engineered bifidobacterium. The idea of using bacteria to treat cancers is very interesting but still challenging in many ways. The most concern would be the safety, such as how to control the bacteria dosage and numbers, and how to monitor the toxins produced from bacteria, and how to remove the excess bacteria after treatments. How would the authors think about these issues? I look forward to see some discussions on these topics.
Are there any other research groups working on the iDPS or bifidobacterium? I would suggest more discussions on others' work and compare with the results from the author.
The author mentioned the nanoparticle size as 1000 nm which is a misleading information. Nanomaterial size is around 100 nm, so nearly 1000 nm sized material is considered as microparticle.
I am particularly interested in the page 12, line 422 where the size of bacteria could be changed by gene engineering. Could the author discuss more?
Many figures are in low resolution, which could be improved. Figure 1b is not complete, so is Figure 2. Figure 4 legend is not in a smaller font.
I agree to publish the manuscript after the revision accordingly. Thanks.
Author Response
Reply to Reviewer 3
Thank you very much for your taking time to review my paper.
I replied to your comment and revised my paper according to your advice.
I hope that you will be satisfied with my revision.
Comments and Suggestions for Authors
1)The authors basically reviewed their own work on the iDPS with gene-engineered bifidobacterium. The idea of using bacteria to treat cancers is very interesting but still challenging in many ways. The most concern would be the safety, such as how to control the bacteria dosage and numbers, and how to monitor the toxins produced from bacteria, and how to remove the excess bacteria after treatments. How would the authors think about these issues? I look forward to see some discussions on these topics.
1)According to your comments, I added the following sentences to Discussion.
Please see the line 530-542
For the clinical trial, bacteria dosage was initiated at less than a Noael (No observed adverse effect level) in the most sensitive dogs. And the dose was gradually increased with the minimum homing dose for rat transplanted cancer as a guide.
Since the Bifidobacterium we use is derived from enterobacteria in human, which is also used as intestinal regulators and yogurt as probiotics, and thus it has been considered to be a priori non-toxic. However, to evaluate the toxicity of genetically modified B. longum, a number of preclinical studies have been also carried out in several animal species, including normal mice, nude mice, normal rats, nude rats, dogs and monkeys. Both pharmacological and preliminary general toxicity studies were done, none of which revealed serious unfavorable toxicities.
To eliminate excess Bifidobacteria after treatment, and many antibiotics were found to be effective. In particular, we have confirmed that it can be easily removed with commonly used penicillin antibiotics.
2) Are there any other research groups working on the iDPS or Bifidobacterium? I would suggest I would suggest more discussions on others' work and compare with the results from the author.
2)The following comments were added to the discussion. Please see the line 519-528
At the preclinical level, there are some studies using Bifidobacteria, though there seems to be only our group that has advanced to clinical trials. One of the preclinical studies similar to ours demonstrated the successful delivery and efficient expression of Tumstatin, a powerful angiostatin, with genetically engineered Bifidobacterium, leading to antitumor effects through inducing apoptosis of tumorous vascular endothelial cells[66]. They observed that Bifidobacterium longum (BL), selectively localizes to and proliferates in the hypoxia location within solid tumor. The other investigated the therapeutic effect of new recombinant Bifidobacterium breve strain expressing interleukin (IL)-24 gene on head and neck tumor xenograft in mice, and reported that new recombinant bacterium has the capability of targeting tumor tissue in vivo[67]. These reports are consistent with our results in terms of selective localization of Bifidobacteium in tumors.
3)The author mentioned the nanoparticle size as 1000 nm which is a misleading information. Nanomaterial size is around 100 nm, so nearly 1000 nm sized material is considered as microparticle. 1000nm
3)Certainly, 1000nm may be too big to be called a nanomaterial size. However, it is based on Maeda's paper that the EPR effect can be expanded to 1000 nm. I cited their sentenses for this issues as follows. Please see the line 99-102
Maeda’s group noted that though EPR effect is applicable to particles of μm size (i.e., bacteria) or macromolecules of ∼1000 kDa, nano carriers with diameters of ∼100 nm are known to achieve better or optimal EPR‐based tumor accumulation[29].
I am particularly interested in the page 12, line 422 where the size of bacteria could be changed by gene engineering. Could the author discuss more?
The following sentences were added to Discussion. Please see the line 603-615.
Tunability of bacterial size has been investigated [78]. Interestingly, it was reported that deletion of the Bacillus subtilis ponA gene encoding for PBP1 (a class A penicillin-binding protein), a bifunctional peptidoglycan synthase, led to thinner cells [79], indicating that it may be possible to change the size of the bacteria by gene engineering.
Many figures are in low resolution, which could be improved. Figure 1b is not complete, so is Figure 2. Figure 4 legend is not in a smaller font.
I reformed fig.1-3 and table1, and changed the font of F-g.2,4 into smaller one.
